# Examination of multilevel domains of minority stress: Implications for drug use and mental and physical health among Latina women who have sex with women and men

Alice Cepeda[1]*, Kathryn M. Nowotny[2], Jessica Frankeberger[3], Esmeralda Ramirez[1], Victoria E. Rodriguez[4], Tasha Perdue[1], Avelardo Valdez[1]

**1** Suzanne Dworak-Peck School of Social Work, University of Southern California, Los Angeles, CA, United States of America, **2** Department of Sociology, University of Miami, Miami, FL, United States of America, **3** Department of Behavioral and Community Health Sciences, Graduate School of Public Health, University of Pittsburgh, Pittsburgh, PA, United States America, **4** Department of Population Health and Disease Prevention, Program in Public Health, University of California, Irvine, Irvine, CA, United States America

* alicecep@usc.edu

## Abstract

There has recently been growing attention and concern in the U.S. on the detrimental drug use and related health conditions impacting diverse sexual minority populations. While some evidence indicates that bisexual women are at increased risk of substance use, little attention has been given to disadvantaged and racial/ethnic minority bisexual women, who are particularly vulnerable to a complexity of stressors and risk. Using data from a 15-year longitudinal study in San Antonio, Texas, the current study examines drug use, incarceration histories, stressful life events, and infections among 206 young adult Mexican-American women who report engaging in sex with both men and women (WSWM) (n = 61) and those indicating having exclusively male sex partners (WSM) (n = 145). A bivariate analysis finds that WSWM experienced more frequent (p = 0.001) and longer total time incarcerated (p = 0.001), as well as exposure to more stressful life events (p = 0.003). WSWM also have higher rates of past 30 day injection drug use (p = 0.026) and related Hepatitis C Virus (HCV) infection (p = 0.001), as well as greater symptomatology associated with depression (p = 0.014), PTSD (p = 0.005), and suicidal ideation (p = 0.036). Findings indicate a significantly elevated risk profile for socio-economically marginalized WSWM. This knowledge is timely and central to policy discourse to develop interventions and health campaigns aimed at reducing and/or preventing further health disparities among this highly susceptible population of minority women.

## Introduction

Bisexual-identified women and women who have sex with women and men (WSWM) have substantially worse health profiles than women who engage in exclusively lesbian and

**Data Availability Statement:** All relevant data are within the manuscript and its Supporting Information files.

**Funding:** This study was funded by the National Institutes of Health, National Institute on Drug Abuse (NIDA) under grant R01DA039269 awarded to AC and AV. The funders had no role in study design, data collection and analysis, decision to publish, or preparation of the manuscript.

**Competing interests:** The authors have declared that no competing interests exists.

heterosexual behaviors [1–3]. Recent data indicate that 20% of U.S. women identify as a sexual minority or report ever having a same-sex sexual relationship [4], and women aged 18–44 are increasingly identifying as bisexual [5]. There has been limited research on the patterns of drug use and associated detrimental physical and mental health conditions among behaviorally bisexual women, especially for race/ethnic minority women [1, 6]. In general, research on sexual minority Latinas is lacking, with research on health disparities for Latina WSWM almost non-existent. This is a critical oversight for this important health disparity population because these women exist in a unique social space at the intersection of gender, race/ethnicity, and sexuality. People with intersecting subordinated statuses, such as ethnic minority women, tend to be "marginal members within marginalized groups" relegating "them to a position of acute social invisibility" [7].

While research has recognized bisexual women's elevated health risks compared to lesbian or heterosexual women for chronic and infectious diseases, obesity, and cardiovascular disease [2, 8–11], less is known about the nature and prevalence of drug use among WSWM, especially among those that do not identify as bisexual [5, 12]. A few studies have consistently shown that lesbian- and bisexual-identified women have higher rates of substance use compared to their heterosexual counterparts [8, 10, 13, 14]. For instance, a recent study found that "mostly heterosexual" women compared to "exclusively heterosexual" women are more likely to report lifetime tobacco, marijuana, and past year alcohol use [15]. Research also documents an increased likelihood of ever injecting drugs among women who have sex with other women [16, 17], as well as more overdoses and higher rates of enrollment into addiction treatment [18]. In contrast, Bostwick and colleagues found relatively few differences between a community-based sample of lesbian and bisexual women's lifetime cocaine and marijuana use [9]. Similarly, German and Latkin found no differences in substance use patterns for lesbian and bisexual women compared to heterosexual women [19].

One of the proposed mechanisms by which same sex sexual behavior is associated with deleterious health outcomes is increased exposure to interpersonal stress and stressful life events. The minority stress model posits that members of highly stigmatized groups experience chronically high levels of stress resulting from discrimination (e.g., overt, internalized), disproportionate exposure to stressful life events (e.g., violent victimization, criminal justice involvement), and low socioeconomic status [20–22]. The burden of these stressors can lead to negative health outcomes, such as higher mortality rates, as well as the use of negative coping behaviors like substance use [20–22]. It is imperative to acknowledge the complex nature of health disparities that span multi-level domains of influence and extend the focus beyond the individual to value the importance of social and structural factors [23, 24].

There is research to suggest that sexual minority women, in general, are negatively impacted by key structural determinants of health, such as involvement with the criminal justice system, un/under employment, and poverty, as well as interpersonal determinants, such as violence and discrimination. For example, a disproportionate number of sexual minority women are incarcerated—42.1% of women in prison and 35.7% in jail [25]. These women not only report harsher punishment while incarcerated, but also have longer sentences [19], greater psychological distress, and histories of sexual victimization [25]. The National Violence Against Women survey found that 35.4% of women reporting a history of cohabitation with a same-sex partner had experienced physical abuse in their lifetimes. The corresponding rate of abuse for women with a history of only opposite-sex cohabitation was 20.4% [26]. Approximately 1 in 5 bisexual women (22.1%) have been raped by an intimate partner, with 40% experiencing sexual violence other than rape [27]. A study of patients at reproductive health clinics found that WSWM were significantly more likely than women who have sex exclusively with men to report a lifetime history of intimate partner violence (IPV), and, after controlling

for IPV, reported higher numbers of sexual risk behaviors, male-perpetrated reproductive coercion, and STIs [2].

Sexual minority status is associated with increased health risks among women of color [28, 29]. In one of the few studies with sexual minority women that included a subsample of Latinas, bisexual Latinas were more likely to report psychological distress than were non-Hispanic white bisexual women [29]. More generally, in Latinx populations, acculturation is associated with increased exposure to discrimination and substance use [13]. Thus, sexual minority Latinx adults are more likely to abuse alcohol than heterosexual Latinx adults [30].

We examine differences between two groups of Latina women—(1) women who report having sex at least once with both men and women (i.e., WSWM) and (2) women who report having sex at least once with men only—within domains of influence at the individual-level (demographic background, acculturation), interpersonal-level (violent victimization, stressful life events), and structural-level (criminal justice involvement, unemployment, unstable housing). We then examine disparities in health including drug use, physical health, and mental health. The study sample includes a cohort of Mexican-American young adult women who live in neighborhoods in San Antonio, TX USA characterized by concentrated poverty, high unemployment, active drug markets, high levels of crime, and high rates of male incarceration and criminal youth gangs [31]. The San Antonio population is more than 1 million, of which more than 50% is of Mexican descent. San Antonio is among the top 10 cities with the largest number of people living in distressed zip codes and has the highest level of spatial inequality between zip codes [32]. The most distressed zip code was the site for the current study. These women occupy multiple, intersecting stigmatized statuses [33] and live in a context of structural disadvantage, which may result in stressors and health outcomes that are likely amplified for women engaging in same sex sexual behaviors.

## Materials and methods

The San Antonio Latina Trajectory Outcomes (Proyecto SALTO) study is an on-going (2015–2020) community-based follow-up study examining the long-term health outcomes of drug use and intimate partner violence (IPV) among a cohort of Mexican American women. The cohort was originally identified and recruited in San Antonio, TX USA during 1999–2001 [31]. Eligibility criteria for Proyecto SALTO and the original study included being a Mexican-American female, being aged 14 to 18 at the time of the original study, and being associated with one of 27 male street gangs from the San Antonio catchment area. The ongoing study employs a concurrent mixed-method (CMM) nested longitudinal cohort design including the collection of biological, survey, and qualitative data. The current study sample consists of a preliminary sample of women who completed survey interviews and biological testing as part of the current follow-up study (n = 206). All study protocols were approved by the Institutional Review Board at University of Southern California and informed consent was obtained from all participants.

### Measures

**Individual-level.** Sexual orientation is typically defined and measured in terms of three dimensions—behavior, attraction, and identity—yet, operationally defining and measuring sexual orientation is a challenge [1]. In our study, the stratifying variable was behavior; specifically reporting same-sex and opposite-sex sexual behaviors compared to only opposite-sex sexual behaviors. The time period for reporting behavior is lifetime. All women in the cohort reported having sex with a man at least once in their lifetime, while only some reported having sex with a woman (women) as well. A subsample of women were asked to self-report their

sexual identity. A measure for sexual identity was not initially included in the follow-up survey because same-sex sexual behaviors had not been reported in initial work done in preparation for the start of the Proyecto SALTO data collection and none of the women reported same-sex partners during adolescence. Once the high prevalence of same-sex sexual behaviors was identified through data monitoring, a new item asking about sexual identity was added to the survey. In the current study sample, 61 out of 206 women (29.6%) reported having sex with both men and women. Overall, 15.8% of women identified as bisexual, 3.8% identified as lesbian, 79.9% identified as heterosexual, and 0.8% identified as other. A comparison of selected demographic and background characteristics show that women who were asked the sexual identity question are not systematically different from women who were not asked the question (see Table A1 in S1 Appendix).

Sociodemographic data was collected on age, years of school completed, marital status, and number of children (including biological, step, and adopted). On average, women were 33.3 years old and had completed 11.0 years of school. Half of women were currently married or cohabitating with a partner, and 15.1% were separated/divorces/widowed with the remaining being single/never married. Women had 3.3 children on average. The Acculturation Rating Scale for Mexican Americans-II was used to measure positive and negative concepts of identity through assessing integration, assimilation, separation, and marginalization (α = 0.83) [34]. Scores were standardized with higher values indicating more Anglo orientation and lower values indicating more Mexican orientation.

**Interpersonal-level.** Childhood trauma was measured using the 28-item Childhood Trauma Questionnaire (CTQ). This instrument assesses abuse and neglect in childhood and adolescence through five clinical scales measuring: emotional abuse (α = 0.84; range 0–24), emotional neglect (α = 0.91; range 0–23), physical abuse (α = 0.87; range 0–25), physical neglect (α = 0.55; range 0–19), and sexual abuse (α = 0.94; range 0–25) [35, 36].

Intimate partner violence (IPV) victimization was measured using the Revised Conflict Tactics Scales (CTS2)[37, 38]. We report four subscales measuring victimization for physical assault (α = 0.81), psychological aggression (α = 0.76), injury (α = 0.84), and sexual coercion (α = 0.76). The scoring shows the prevalence of women who experienced different types of IPV with their current partner during the past year (163 women out of 206 reported having a partner). We also report a measure of chronicity, with higher numbers representing more frequent victimization, and severity, with items categorized as "minor" victimization and "severe" victimization. According to Straus and Douglas [38], examples of "severe" forms of sexual coercion were: "My partner used force (like hitting, holding down, or using a weapon) to make me have sex" and "My partner used threats to make me have sex." An example of "minor" sexual coercion was "My partner insisted on sex when I did not want to (but did not use physical force)." Overall, women with current romantic partners reported alarming levels of IPV during the previous 12 months: 86.4% psychological aggression, 77.3% physical assault, 69.9% sexual coercion, and 74.9% injury. When looking at only "severe" forms of IPV, the prevalence rates are only slightly reduced: 67.5% psychological aggression, 77.3% physical assault, 65.0% sexual coercion, and 39.6% injury.

Stressful life experiences were measured using the 13-item self-report Stressful Life Events Screening Questionnaire (SLESQ) [39]. The SLESQ assesses lifetime exposure to eleven specific and two general categories of traumatic events meeting Criterion A1 of PTSD in the DSM-IV [40], such as a life-threatening accident, physical and sexual abuse, and witness to another person being killed or assaulted. The mean number of stress life events experienced is 2.82.

**Structural-level.** Structural-level determinants included differential involvement with key societal institutions, such as involvement with Child Protective Services (CPS) and

incarceration history, including ever incarcerated, number of times incarcerated for 30 days or longer (0 times, 1 time, 2+ times), and total number of years incarcerated. Overall, one-third of mothers reported some type of CPS involvement and two-thirds of women overall reported being incarcerated at least once in their lifetime. Dichotomous measures of unemployment (34.9%) and unstable housing (22.5%; including living most of the previous year in a halfway house, motel, jail, or three or more locations) were also examined.

**Health outcomes.**　A number of drug use outcomes were assessed. Self-reported lifetime drug use was collected for marijuana, sedative/hypnotics, methamphetamines, crack/cocaine, non-injecting heroin, prescription opioids, injection drug use, and methadone. Lifetime treatment for drug use was measured using the question, "Have you ever sought treatment for problems with your drug or alcohol use?" Lifetime drug dependence for marijuana ($\alpha = 0.85$), cocaine ($\alpha = 0.85$), methamphetamines ($\alpha = 0.85$), and opioids ($\alpha = 0.85$) was measured using the 5-item Severity of Dependence Scale (SDS) [41]. For this summative scale, a score of 3 + indicates a likely diagnosis of substance dependence according to the DSM [41–43]. However, we used more precise and conservative cut-off scores that have been identified for specific types of drug use: a score of 4+ for marijuana [44], 3+ for cocaine [45], 4+ for methamphetamine [46], and 5+ for opioids [47]. Using this criteria, about 60% of women, overall, reported lifetime drug dependence (60.6% marijuana, 65.3% cocaine, 60.0% methamphetamine, and 57.8% opioids). Current drug use was assessed by measuring self-report past 30-day injecting drug use and drug metabolite testing using the iCup AD 8 Panel urine test from Alere Toxicology. Positive urinalysis was documented at the following rates: 23.8% marijuana, 12.9% cocaine, 16.8% methamphetamines, and 23.9% opioids. Nine percent of women reported injecting drugs in the past month.

Physical health included clinical testing for STI and other infectious diseases. Specimens were tested using HIV antibody using enzyme linked immunoassay (EIA); Herpes Simplex Virus-Type 2 (HSV-2) specific IgG antibody test with an index ratio > 0.9 (HerpeSelect HSV-2 ELISA, Focus Technologies); and Hepatitis C Virus (HCV) antibody assays using Abbott HCV EIA 3.0 procedure for encoded antigens (recombinant c100-3, HC-31, and HC-34). A positive HCV antibody test may indicate a cleared or ongoing chronic infection. We were unable to conduct HCV RNA tests to confirm current infection. The tested prevalence for HCV is 27.8% (49/176). Only two women tested positive for HIV, so this measure was not included in the analysis (both women knew their status prior to enrolling in the study). Not all women were able to provide blood and or urine samples for clinical testing due to not being able to finish the interview, collapsed veins, etc. The tested prevalence rate for herpes is 62.4% (106/170), 5.7% for chlamydia (10/175), and 3.3% for gonorrhea (6/175).

Metabolic syndrome is a composite measure of risk for heart disease and other health problems such as stroke and diabetes. Women were considered high-risk if they were within the "risk" categories for three or more of the following: BMI, triglycerides, HDL cholesterol, blood pressure, and blood glucose. Overall, 61.5% of women met the criteria for obesity, 39.2% for high triglycerides, 20.0% for low HDL cholesterol, 70.0% for high blood pressure, and 36.3% for high blood glucose. BMI and blood pressure were assessed by study staff and the remaining outcomes were assessed through clinical blood draws. A standard 4-point self-rated health question was used to assess poor/fair health: "How would you describe your health status? Would you say it is. . . excellent, good, fair, or poor?" was used for self-reported health (0 = good or excellent, 1 = fair or poor). Two-thirds of women rated their health as fair or poor (66.0%).

Mental health measures included depression, post-traumatic stress disorder (PTSD), and suicidal ideation. Depressive symptomology ($\alpha = 0.92$) was measured using the eight-item version of the Center for Epidemiological Studies Depression Scale (CES-D) [48]. The CES-D

ranges from 0–24 with higher scores indicating depressive symptomology, but not a clinical diagnosis of depression [49]. Mean symptomatology is 9.88 with 25.2% above the recommended threshold for depression ($\geq$ 16). The 17-item PTSD Checklist-Civilian Version was used to measure PTSD symptoms with higher scores indicating higher levels of symptom severity ($\alpha$ = 0.95) [50]. The mean symptomatology is 23.8 with 54.9% of women meeting the recommended criteria for PTSD. A positive response to one of four questions from the General Health Questionnaire [51]: have you recently "found that the idea of taking your own life kept coming into your mind"; "found yourself wishing you were dead and away from it all"; "thought of the possibility that you might make away with yourself"; and "felt that life isn't worth living" were used to measure suicidal ideation. Twenty-one percent of women reported suicidal ideation in the past year.

## Analysis

The analysis used bivariate tests of association (two-tailed t-test and chi square test of independence) to examine differences in exposure to potential stressors among WSWM and WSM at the individual-level (demographic background, acculturation), interpersonal-level (violent victimization, stressful life events), and structural-level (criminal justice involvement, unemployment, unstable housing). We then examine disparities in health including drug use, physical health, and mental health. Effect sizes were calculated for continuous variables using Cohen's d with z-score transformed variables and bootstrapped standard errors reported. The effect size was standardized to make comparisons across variables with different ranges of measurement [52]. The effect size for categorical variables was reported as binary odds ratios (OR).

## Results

### Individual level

As shown in Table 1, there were no individual-level differences between WSWM and WSM with regards to age, education, number of children, housing, and marital status. Differences in acculturation trended towards significance (p = 0.052) with mean acculturation scores for WSM (0.57) higher than WSWM (0.31). More than half (53.9%) of WSWM reported identifying as non-heterosexual compared to 6.7% of WSM in the study (OR = 16.33, p = 0.001). Of WSWM, 46.2% identified as heterosexual, 12.8% as lesbian, 38.5% as bisexual, and 2.6% as

**Table 1. Differences in individual-level factors between Women who have Sex with Men (WSM) and Women who have Sex with Women and Men (WSWM) in proyecto SALTO.**

| | | WSM (n = 145) | | | | WSWM (n = 61) | | | | | | |
|---|---|---|---|---|---|---|---|---|---|---|---|---|
| | n | Mean | se | % | n | Mean | se | % | t/x² | p | d/OR (se) | p |
| Age | 145 | 33.43 | 0.18 | | 61 | 32.87 | 0.34 | | 1.58 | .115 | 0.24 (0.18) | 0.186 |
| Acculturation | 137 | 0.57 | 0.07 | | 54 | 0.31 | 0.10 | | 2.09 | .038 | 0.34 (0.17) | 0.052 |
| Years of Education | 145 | 11.11 | 0.17 | | 61 | 10.59 | 0.25 | | 1.69 | .092 | 0.26 (0.15) | 0.095 |
| Marital Status | | | | | | | | | 0.27 | .874 | | |
| Single, Never Married | 55 | | | 37.9 | 21 | | | 34.4 | | | 0.86 (0.27) | 0.634 |
| Separated/Divorced/Widowed | 21 | | | 14.5 | 10 | | | 16.4 | | | 1.16 (0.49) | 0.726 |
| Married/Cohabitating | 69 | | | 47.6 | 30 | | | 49.2 | | | 1.07 (0.33) | 0.834 |
| Number of Children | 145 | 3.35 | 0.15 | | 61 | 3.18 | 0.22 | | 0.63 | .527 | 0.10 (0.16) | 0.535 |
| Identify as non-Heterosexual[a] | 6 | | | 6.7 | 21 | | | 53.85 | 36.60 | .001 | 16.33 (8.67) | 0.001 |

[a]Based on a reduced sample (n = 129; 90, 39)

other. Almost all WSM identified as heterosexual (93.3%), with 6.7% identifying as bisexual and no WSM (0.0%) identifying as lesbian or other. Fifteen out of 61 WSWM (24.6%) reported having a primary sexual partner who was a woman during the past year. There were 39 WSWM (out of 61) that were asked to report their sexual identity (see Methods). Seven out of the 39 WSWM reported having a primary sex partner who was a woman in the past year. Among those seven, one identified as heterosexual, four identified as lesbian, and two identified as bisexual. Five WSWM reported having a non-primary sex partner who was a woman during the same time period. Three of these women were asked the sexual identity question and all identified as bisexual.

## Interpersonal level

Mean scores for childhood emotional abuse (d = 0.36, p = 0.031) and stressful life events (d = 0.47 p = 0.003) were higher among WSWM compared to WSM (Table 2). On average, WSWM reported 3.56 stressful life events, while WSM reported 2.65. There were no differences in past-year prevalence of IPV between WSWM and WSM. However, among women experiencing past-year IPV, WSWM experienced psychological aggression (d = 0.38, p = 0.047) and physical assaults (d = 0.39, p = 0.041) at a higher frequency. The effect sizes suggest that this is a non-trivial difference. We also examined severity of IPV among women who experienced IPV in the past year. Almost all women (> 80%) in both groups reported "severe"—as opposed to "only minor"—forms of psychological aggression, physical assault, and sexual coercion. Half of WSWM reported "severe" injury from their current partner during the past year, compared to 31.0% of WSM (OR = 2.23, p = 0.046). Finally, WSWM were also more likely to report ever being forced to engage in sex work (16.5%) compared to 2.8% of WSM (OR = 6.91, p = 0.002).

## Structural level

There were no structural differences related to employment, lifetime involvement with CPS, or housing between WSWM and WSM (Table 2). Overall, lifetime incarceration rates for these women far exceeded national rates. About half of WSM (54.5%) have been incarcerated at least once, compared to almost all of WSWM (83.6%). WSWM had over four times the odds of experiencing lifetime incarceration (OR = 4.26, p = 0.001). When examining substantive incarceration episodes, we found that 47.5% of WSWM experienced two or more incarceration episodes lasting 30 days or longer, compared to 18.6% of WSM (OR = 3.96, p = 0.001). In total, WSWM have spent an average of 1.83 years behind bars. The most common charges reported were drug related (i.e., possession, intent to distribute). Still, 14 women have been incarcerated for prostitution, 10 of whom reported engaging in same-sex sexual behavior (16.4% vs. 2.8%, OR = 6.91, p = 0.002), with the number of incarceration episodes for prostitution ranging from one to five. WSWM also have significantly higher rates of having a felony conviction (50.8% vs. 29.0%, OR = 2.53, p = 0.003).

## Health outcomes: Drug use

Higher rates of drug use were observed among WSWM (Table 3). Over one-third of WSWM (35.0%) tested positive for opioids compared to 19.0% of WSM (OR = 2.30, p = 0.017), with WSWM more likely to report injecting drug use during the past 30 days (16.4% vs 6.2%, OR = 2.96, p = 0.026). These women also reported higher lifetime prevalence of sedatives/ hypnotics, crack/cocaine, methadone, noninjecting heroin use, prescription opioids, and injecting drug use. Drug treatment utilization was higher for WSWM (49.2%), compared to 30.3% of WSM (OR = 2.22, p = 0.011). WSWM were more likely than WSM to meet the criteria for

**Table 2. Differences in interpersonal-level and structural-level factors between Women who have Sex with Men (WSM) and Women who have Sex with Women and Men (WSWM) in proyecto SALTO.**

| | WSM (n = 145) | | | | WSWM (n = 61) | | | | | | | |
|---|---|---|---|---|---|---|---|---|---|---|---|---|
| | n | Mean | se | % | n | Mean | se | % | t/x$^2$ | p | d/OR (se) | p |
| Interpersonal-Level Factors | | | | | | | | | | | | |
| Childhood Trauma | | | | | | | | | | | | |
| Emotional Abuse | 145 | 8.50 | 0.44 | | 61 | 10.50 | 0.80 | | 2.35 | 0.020 | 0.36 (0.17) | 0.031 |
| Physical Abuse | 145 | 5.98 | 0.38 | | 61 | 7.62 | 0.81 | | 2.09 | 0.038 | 0.32 (0.17) | 0.056 |
| Sexual Abuse | 145 | 8.37 | 0.57 | | 61 | 8.74 | 0.89 | | 0.36 | 0.723 | 0.05 (0.14) | 0.708 |
| Emotional Neglect | 145 | 11.56 | 0.38 | | 61 | 12.43 | 0.61 | | 1.24 | 0.218 | 0.19 (0.15) | 0.216 |
| Physical Neglect | 145 | 11.11 | 0.22 | | 61 | 10.97 | 0.36 | | 0.34 | 0.736 | 0.05 (0.14) | 0.722 |
| Past Year Prevalence of IPV[a] | | | | | | | | | | | | |
| Psychological Aggression | 94 | | | 81.7 | 41 | | | 85.4 | 0.32 | 0.570 | 1.31 (0.62) | 0.571 |
| Physical Assault | 88 | | | 76.5 | 38 | | | 79.2 | 0.14 | 0.713 | 1.17 (0.49) | 0.713 |
| Sexual Coercion/ Rape | 77 | | | 67.0 | 37 | | | 77.1 | 1.65 | 0.199 | 1.66 (0.66) | 0.201 |
| Injury | 84 | | | 73.0 | 38 | | | 79.2 | 0.67 | 0.412 | 1.40 (0.58) | 0.413 |
| Frequency of IPV[b] | | | | | | | | | | | | |
| Psychological Aggression | 94 | 10.77 | 0.51 | | 41 | 13.05 | 1.21 | | 2.07 | 0.041 | 0.38 (0.19) | 0.047 |
| Physical Assault | 88 | 12.14 | 0.64 | | 38 | 15.21 | 1.80 | | 2.01 | 0.047 | 0.39 (0.19) | 0.041 |
| Sexual Coercion/ Rape | 77 | 6.69 | 0.49 | | 37 | 8.27 | 1.10 | | 1.52 | 0.131 | 0.30 (0.23) | 0.191 |
| Injury | 84 | 7.61 | 0.59 | | 38 | 7.74 | 0.79 | | 0.13 | 0.899 | 0.03 (0.19) | 0.897 |
| Forced Sex Work | 4 | | | 2.8 | 10 | | | 16.4 | 12.60 | 0.001 | 6.91 (4.24) | 0.002 |
| Stressful Life Events | 145 | 2.65 | 0.15 | | 61 | 3.56 | 0.30 | | 3.06 | 0.003 | 0.47 (0.16) | 0.003 |
| Structural-Level Factors | | | | | | | | | | | | |
| Unemployed/ Occasional | | | | 32.4 | | | | 39.3 | 0.91 | 0.339 | 1.35 (0.43) | 0.340 |
| Employment | 47 | | | | 24 | | | | | | | |
| Lifetime Involvement with CPS[c] | 43 | | | 31.9 | 24 | | | 41.4 | 1.63 | 0.202 | 1.51 (0.50) | 0.204 |
| Unstable Housing | 27 | | | 18.6 | 19 | | | 31.2 | 3.89 | 0.049 | 1.98 (0.69) | 0.051 |
| Incarceration History | | | | | | | | | | | | |
| Ever Incarcerated (Y/N) | 79 | | | 54.5 | 51 | | | 83.6 | 15.64 | 0.001 | 4.26 (1.64) | 0.001 |
| Incarcerated for 30+ Days | | | | | | | | | 24.78 | 0.001 | | |
| 0 times | 103 | | | 71.0 | 21 | | | 34.4 | | | 0.21 (0.07) | 0.001 |
| 1 time | 15 | | | 10.3 | 11 | | | 18.0 | | | 1.91 (0.82) | 0.134 |
| 2+ times | 27 | | | 18.6 | 29 | | | 47.5 | | | 3.96 (1.32) | 0.001 |
| Total Years Incarcerated | 145 | 0.50 | 0.11 | | 61 | 1.83 | 0.34 | | 4.84 | 0.001 | 0.74 (0.17) | 0.001 |
| Felony Conviction | 42 | | | 29.0 | 31 | | | 50.8 | 8.96 | 0.003 | 2.53 (0.80) | 0.003 |

[a]Among women reporting a current romantic relationship (n = 163; 115, 48). IPV = intimate partner violence

[b]Among women who experienced intimate partner violence.

[c]Among mothers (n = 193; 135, 58); CPS = Child Protective Services

lifetime drug dependence for marijuana (73.8% vs 57.9%, OR = 2.04, p = 0.034) and for heroin/opioids (70.5% vs. 55.2%, OR = 1.94, p = 0.042).

## Health outcomes: Physical and mental health

The preliminary findings also indicate that that WSWM have higher risks for some physical and mental health conditions (Table 4). While there were no significant differences in STI prevalence or self-rated poor health, WSWM had a tested prevalence of HCV that was over two times higher than WSM (44.6% vs. 20.5%, OR = 3.13, p = 0.001). WSWM had higher

**Table 3. Differences in drug use between Women who have Sex with Men (WSM) and Women who have Sex with Women and Men (WSWM) in proyecto SALTO.**

| | WSM (n = 145) | | WSWM (n = 61) | | | | | |
| --- | --- | --- | --- | --- | --- | --- | --- | --- |
| | n | % | n | % | $x^2$ | p | OR (se) | p |
| Lifetime Drug Use | | | | | | | | |
| Marijuana | 129 | 89.0 | 56 | 91.8 | 0.38 | 0.539 | 1.39 (0.75) | 0.540 |
| Sedatives/Hypnotics | 45 | 31.7 | 28 | 46.7 | 4.09 | 0.043 | 1.89 (0.60) | 0.044 |
| Methamphetamines | 34 | 29.7 | 25 | 43.3 | 3.56 | 0.059 | 1.81 (0.58) | 0.061 |
| Crack/Cocaine | 103 | 71.0 | 52 | 85.3 | 4.66 | 0.031 | 2.36 (0.95) | 0.034 |
| Non-Injecting Heroin | 54 | 37.5 | 35 | 58.3 | 7.47 | 0.006 | 2.33 (0.73) | 0.007 |
| Prescription Opioids | 22 | 15.2 | 19 | 31.2 | 6.87 | 0.009 | 2.53 (0.91) | 0.010 |
| Injection Drug Use | 37 | 25.7 | 30 | 49.2 | 10.74 | 0.001 | 2.80 (0.89) | 0.001 |
| Methadone | 43 | 23.5 | 26 | 41.0 | 6.46 | 0.011 | 2.27 (0.74) | 0.012 |
| Lifetime Treatment for Drug Use | 44 | 30.3 | 30 | 49.2 | 6.62 | 0.010 | 2.22 (0.70) | 0.011 |
| Lifetime Drug Dependence | | | | | | | | |
| Marijuana | 84 | 57.9 | 45 | 73.8 | 4.60 | 0.032 | 2.04 (0.69) | 0.034 |
| Cocaine/Crack | 92 | 63.5 | 47 | 77.1 | 3.62 | 0.057 | 1.93 (0.68) | 0.059 |
| Methamphetamines | 84 | 57.9 | 44 | 72.1 | 3.68 | 0.055 | 1.88 (0.62) | 0.057 |
| Heroin/Opioids | 80 | 55.2 | 43 | 70.5 | 4.19 | 0.041 | 1.94 (0.63) | 0.042 |
| Currernt Drug Use | | | | | | | | |
| Injecting Drug Use Past 30 Days | 9 | 6.2 | 10 | 16.4 | 5.32 | 0.021 | 2.96 (1.45) | 0.026 |
| Urinalysis Results[a] | | | | | | | | |
| Marijuana | 29 | 21.0 | 19 | 31.7 | 2.58 | 0.108 | 1.74 (0.61) | 0.110 |
| Cocaine | 20 | 14.5 | 5 | 8.3 | 1.44 | 0.230 | 0.53 (0.28) | 0.236 |
| Methamphetamine | 21 | 15.2 | 11 | 18.3 | 0.30 | 0.584 | 1.25 (0.51) | 0.585 |
| Opioids | 26 | 19.0 | 21 | 35.0 | 5.90 | 0.015 | 2.30 (0.80) | 0.017 |

[a]Based on reduced sample size n = 198 (138, 60).

mean symptomatology, compared to WSM, for depression (d = 0.37, p = 0.015) and PTSD (d = 0.41, p = 0.008). Using the suggested clinical threshold, 72.1% of WSWM met the DSM criteria for PTSD, with 2.55 higher odds (p = 0.005) compared to WSM (50.3%). The

**Table 4. Differences in physical health and mental health between Women who have Sex with Men (WSM) and Women who have Sex with Women and Men (WSWM) in proyecto SALTO.**

| | WSM (n = 145) | | | | WSWM (n = 61) | | | | | | | |
| --- | --- | --- | --- | --- | --- | --- | --- | --- | --- | --- | --- | --- |
| | n | Mean | se | % | n | Mean | se | % | $t/x^2$ | p | d/OR (se) | p |
| Physical Health | | | | | | | | | | | | |
| Sexually Transmitted Infections (Tested) | | | | | | | | | | | | |
| Herpes Simplex Virus—Type 2 | 67 | | | 58.8 | 38 | | | 71.7 | 2.59 | 0.108 | 1.78 (0.64) | 0.110 |
| Gonorrhea | 3 | | | 2.6 | 2 | | | 3.6 | 0.15 | 0.696 | 1.43 (1.33) | 0.698 |
| Chlamydia | 6 | | | 5.2 | 4 | | | 7.3 | 0.31 | 0.575 | 1.45 (0.97) | 0.577 |
| Hepatitis C (Tested) | 24 | | | 20.5 | 25 | | | 44.6 | 10.89 | 0.001 | 3.13 (1.10) | 0.001 |
| Metabolic Syndrome (Tested) | 87 | | | 60.0 | 27 | | | 44.3 | 4.30 | 0.038 | 0.53 (0.16) | 0.039 |
| Self-Rated Poor Health | 92 | | | 66.2 | 36 | | | 66.7 | 0.01 | 0.950 | 1.02 (0.35) | 0.950 |
| Mental Health | | | | | | | | | | | | |
| Depressive Symptoms | 145 | 9.40 | 0.58 | | 61 | 12.02 | 0.87 | | 2466 | 0.015 | 0.38 (0.15) | 0.014 |
| PTSD Symptoms | 145 | 22.31 | 1.58 | | 61 | 30.03 | 2.29 | | 2.71 | 0.007 | 0.41 (0.15) | 0.005 |
| Suicidal Ideation | 22 | | | 15.2 | 17 | | | 27.9 | 4.51 | 0.034 | 2.16 (0.79) | 0.036 |

Herpes n = 167 (114, 53). Gonorrhea/Chlamydia n = 172 (117, 55). HCV n = 173 (117, 56)

prevalence of depression was not significantly different between the two groups of women (29.5% vs. 23.5%, respectively), but suicidal ideation was higher among WSWM (27.9% vs. 15.2%, OR = 2.16, p = 0.036).

## Discussion

The findings provide evidence for a health syndemic [53] for socio-economically marginalized Latina WSWM, including exposure to multi-level domains of risk. For both WSWM and WSM in this sample, the contexts in which they live create conditions that favor the transmission of diseases, as evidenced by the overall high rates of STI and HCV. Women also appear to live in conditions that increase drug use—more so for WSWM, which is consistent with some of the previous research pointing to bisexual women's increased risk for drug use and health outcomes compared to heterosexual women [10, 54, 55]. Of particular concern are the high rates of opioid use, including prescription pills, noninjecting heroin, and injecting heroin. These drug use patterns reveal these women's disproportionate susceptibility for a multiplicity of health risks (e.g. overdoses) and poor health outcomes when compared to more conventional population samples. For example, over 70% of WSWM, and 50% of WSM, met the suggested threshold for a current diagnosis of PTSD, with one-quarter of WSWM reporting suicidal ideation. The women in our study are marginalized within an already marginalized group: Mexican-American middle class women living in the United States [7]. For women that report same sex sexual relationships, regardless of their sexual identity, this marginalization is increased.

Our findings provide support for the minority stress model, which argues that marginalized people experience greater exposure to social stressors such as interpersonal and structural discrimination. The women in this sample are not representative of all Mexican-American women in the United States. Rather, they represent Mexican American women living in disadvantaged urban communities across Texas and the southwest. The reported rates of childhood trauma, IPV, and incarceration history, overall, far exceed national rates among American women of 30%, 22.3%, and 81 to 82 per 100,000 female U.S. residents age 18 and older, respectively, and are more representative of other marginalized women [56–58]. Still, we documented important disparities between WSWM and WSM within interpersonal and structural domains of risk, most notably interpersonal violence and involvement with the criminal justice system.

Incarceration is increasingly being recognized as an important structural factor that shapes race/ethnic disparities in the United States [59, 60]. For Latinx groups, there are additional threats to personal wellbeing due to increasing "crimmigration" [61]. Latinx people simultaneously occupy multiple racialized legal statuses, which increase their chances for criminal justice involvement [62]. For Latinas, gendered expectations surrounding motherhood contribute to additional stigma, which has detrimental consequences for women who use drugs or become involved with the criminal justice system [31, 63]. The disproportionate burden of incarceration and felony convictions among WSWM found in this study, similar to drug use, is likely both a cause and a consequence of the social stressors that they face in their everyday lives. Incarceration itself can be characterized as both a primary and secondary stressor [64], as well as an outcome of racialized policy surrounding drug use and other behaviors [62]. Overall, our findings are in line with previous research documenting the disproportionate number of sexual and gender minorities that are incarcerated [22].

Interpersonal violence is syndemic with drug use and infectious diseases for women in the United States [53, 65], and these intersecting phenomenon are often pathways to incarceration for the women that experience them [66]. The rates of IPV documented in the study sample

are similar to those documented among women in prisons and jails [67]. The majority of women who were in a romantic relationship at the time of the study interview reported experiencing some form of IPV in the past year, including psychological aggression, physical assault, sexual coercion, and injury. We found that WSWM reported experiencing some forms of IPV more often than their WSM counterparts and were more likely to have been severely injured by an intimate partner in the past year. This is in contrast to findings from national surveys, such as the National Violence Against Women Survey, which reports not only lower prevalence rates, in general, but also a starker disparity between heterosexual women and women who have sex with women. The small study sample size may have precluded us from documenting existing disparities between WSWM and WSM, given the overall extreme concentration of violence in this community.

To our knowledge, this is one of the first studies that begins to understand the health status and drug use patterns of Latinas who may or may not identify as sexual minorities, but do engage in same-sex sexual behaviors. While Proyecto SALTO's focus was not initially on sexual minorities, the findings presented here have important time sensitive implications for identifying unique sexual and gender minority populations that often go unnoticed in programs aimed at reducing health disparities. Future data collection and analyses for Proyecto SALTO will focus on documenting the potentially different trajectories of drug use and IPV over the life course of WSWM and WSM, including dynamic models with partner characteristics. This knowledge is central for policy discourse to develop interventions and health campaigns to reduce and/or prevent further disparities among this multiply marginalized group of women.

## Supporting information

**S1 Appendix. Missing on sexual identity.**
(DOCX)

## Acknowledgments

The authors are grateful to Adriana Trevino and Ernestina Cazares for their work on this project.

## Author Contributions

**Conceptualization:** Alice Cepeda, Kathryn M. Nowotny, Jessica Frankeberger, Esmeralda Ramirez, Avelardo Valdez.

**Data curation:** Jessica Frankeberger.

**Formal analysis:** Kathryn M. Nowotny.

**Funding acquisition:** Alice Cepeda, Jessica Frankeberger, Avelardo Valdez.

**Investigation:** Alice Cepeda, Jessica Frankeberger, Esmeralda Ramirez, Avelardo Valdez.

**Methodology:** Alice Cepeda, Kathryn M. Nowotny, Jessica Frankeberger, Avelardo Valdez.

**Project administration:** Alice Cepeda, Jessica Frankeberger, Esmeralda Ramirez, Tasha Perdue.

**Resources:** Alice Cepeda, Kathryn M. Nowotny, Jessica Frankeberger, Esmeralda Ramirez, Victoria E. Rodriguez, Tasha Perdue.

**Supervision:** Alice Cepeda, Esmeralda Ramirez, Avelardo Valdez.

**Visualization:** Alice Cepeda, Kathryn M. Nowotny, Jessica Frankeberger, Victoria E. Rodriguez, Avelardo Valdez.

**Writing – original draft:** Alice Cepeda, Kathryn M. Nowotny, Jessica Frankeberger, Esmeralda Ramirez, Victoria E. Rodriguez, Avelardo Valdez.

**Writing – review & editing:** Alice Cepeda, Kathryn M. Nowotny, Tasha Perdue.

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
