## [Decision Letter · Decision Letter 0]

5 Dec 2019

PONE-D-19-19530

Documenting disparities in drug use, physical health, and mental health for women who have sex with women and men using a cohort of Mexican-American women living in a disadvantaged community

PLOS ONE

Dear Dr. Cepeda,

Thank you for submitting your manuscript to PLOS ONE. After careful consideration, we feel that it has merit but does not fully meet PLOS ONE’s publication criteria as it currently stands. Therefore, we invite you to submit a revised version of the manuscript that addresses the points raised during the review process.

We would appreciate receiving your revised manuscript by Jan 19 2020 11:59PM. To enhance the reproducibility of your results, we recommend that if applicable you deposit your laboratory protocols in protocols.io, where a protocol can be assigned its own identifier (DOI) such that it can be cited independently in the future. For instructions see: http://journals.plos.org/plosone/s/submission-guidelines#loc-laboratory-protocols

We look forward to receiving your revised manuscript.

Kind regards,

Geilson Lima Santana, M.D., Ph.D.

Academic Editor

PLOS ONE

Journal Requirements:

Additional Editor Comments (if provided):

Congratulations for the article.

Please, take into consideration the reviewers´ comments. They will enrich your article for sure.

It is important also to elaborate on the data availability statement. In my opinion, data is not fully present in the text and supplementary material. You can add the databank to your submission. Alternatively, you can make it available in a public repository.

If you don´t have permission to allow public access to it, it is important to state one of the two sentences below:

Data cannot be shared publicly because of [XXX]. Data are available from the XXX Institutional Data Access / Ethics Committee (contact via XXX) for researchers who meet the criteria for access to confidential data.

The data underlying the results presented in the study are available from (include the name of the third party and contact information or URL).

Reviewers' comments:

Reviewer's Responses to Questions

**Comments to the Author**

1. Is the manuscript technically sound, and do the data support the conclusions?

Reviewer #1: Yes

Reviewer #2: Yes

2. Has the statistical analysis been performed appropriately and rigorously? 

Reviewer #1: I Don't Know

Reviewer #2: Yes

3. Have the authors made all data underlying the findings in their manuscript fully available?

Reviewer #1: Yes

Reviewer #2: No

4. Is the manuscript presented in an intelligible fashion and written in standard English?

Reviewer #1: Yes

Reviewer #2: Yes

5. Review Comments to the Author

Reviewer #1: It seems understandable the incomplete availability of data, as this work is part of a larger ongoing project.

All that was left was to better conceptualize, at the beginning of the work, what would be WSWM?, that is, how many episodes of sex with women would count to be classified as WSWM?

As small details:

1. the title is not clear enough

2. reference 5 has some errors

3. DSM IV reference missing

I would like to congratulate the authors, it is an interesting and important work. Both, the structure of the article, including the technical aspects, as well as the content are very good.

Reviewer #2: This is a very interesting and needed article that calls for attention to several key factors that are associated to drug use, physical health and mental health among a sexual minority population - in this case, women who have sex with women and men (WSWM).

Despite being a very specific and vulnerable population, the study results are very interesting and shed light for a poorly understood question. In my opinion, the article could be easily improved if the points listed below are answered/tackled:

1. Would it be possible to adjust the title, possibly making it more attractive? The data shown cover so many different aspects that should not to be simplified as just the "documentation of disparities". Moreover, the socioeconomic vulnerability to which they are exposed (e.g., violence and incarceration) should be another point to be highlighted, instead of just being the context of "living in a disadvantaged community".

2. Include the explanation for not assessing data of women who have sex with women/lesbians in the abstract. It is actually a little bit confusing in the article itself (last paragraph of page 6 and first one of page 7) - since 3.8% answered about their sexual identity as being lesbians, it would not be expected that absolutely 'all women in the cohort reported having sex with men'. Would it be specific of such a population? Moreover, this issue ends up getting more complicated when the results for the sexual identification are observed (lines 251-257) - could you please try to simplify or improve the intelligibility of these excerpts?

3. Please verify the name for 'substance dependence severity scale (SDS)' vs. 'Severity of Dependence Scale (SDS)' (line 188). It would be interesting if authors could explain the reason why they used more conservative cut-off scores as suggested by other researchers - e.g., were the populations similar to the one assessed?

4. In the discussion section, since it is a very particular population, it would be nice to have national rates of the individual-level, interpersonal-level, structural-level and health outcomes that are assessed, or at least identifying those that are rather different from the national level and actually citing the literature rate. For example, line 328 of the article states that the reported rates of childhood trauma, IPV, and incarceration history, overall, far exceed national rates - which are.... I believe this would further show the importance of the present findings and also of how relevant it is to study minority populations like these.

6. PLOS authors have the option to publish the peer review history of their article (what does this mean?). If published, this will include your full peer review and any attached files.

Reviewer #1: Yes: Patricia B Hochgraf

Reviewer #2: No

---

## [Author Response · Author response to Decision Letter 0]

21 Jan 2020

Reviewer #1: It seems understandable the incomplete availability of data, as this work is part of a larger ongoing project.

We have a no-cost extension for the study so technically it has not ended.. Per NIH policy and PLOS ONE’s data policy, the data that support the findings will be made openly available but is not yet ready to be released.

All that was left was to better conceptualize, at the beginning of the work, what would be WSWM?, that is, how many episodes of sex with women would count to be classified as WSWM?

For participants in our study, being labeled a “WSWM” is based on lifetime reports (so having sex with one man and one women at least one time). We believed that this is well clarified in the methods section but have made it more explicit in the final paragraph of the introduction. 

As small details:

1. the title is not clear enough

Reviewer 2 made a similar comment. Please see our response below. 

2. reference 5 has some errors

Thanks for noticing this. We added the sponsor agency of the report to the citation (National Health Statistics Reports, Centers for Disease Control and Prevention). 

3. DSM IV reference missing

The reference has been added to the citation list and the in-text citation has been updated. 

I would like to congratulate the authors, it is an interesting and important work. Both, the structure of the article, including the technical aspects, as well as the content are very good.

Thank you for these kind words. We are excited to share these findings!

Reviewer #2: This is a very interesting and needed article that calls for attention to several key factors that are associated to drug use, physical health and mental health among a sexual minority population - in this case, women who have sex with women and men (WSWM).

Despite being a very specific and vulnerable population, the study results are very interesting and shed light for a poorly understood question. In my opinion, the article could be easily improved if the points listed below are answered/tackled:

1. Would it be possible to adjust the title, possibly making it more attractive? The data shown cover so many different aspects that should not to be simplified as just the "documentation of disparities". Moreover, the socioeconomic vulnerability to which they are exposed (e.g., violence and incarceration) should be another point to be highlighted, instead of just being the context of "living in a disadvantaged community".

Both Reviewers 1 and 2 commented on the title. Following Reviewer 2’s suggestion, we have simplified it to the following: “Documentation of Disparities for Women Who Have Sex With Women and Men Using a Cohort of Mexican-American Women.”

2. Include the explanation for not assessing data of women who have sex with women/lesbians in the abstract. It is actually a little bit confusing in the article itself (last paragraph of page 6 and first one of page 7) - since 3.8% answered about their sexual identity as being lesbians, it would not be expected that absolutely 'all women in the cohort reported having sex with men'. Would it be specific of such a population? Moreover, this issue ends up getting more complicated when the results for the sexual identification are observed (lines 251-257) - could you please try to simplify or improve the intelligibility of these excerpts?

We agree that this is very complex. In our sample, it is correct that all women reported having sex with men independent of their sexual identity. So all of the women in our sample that are lesbian reported having sex with a man at least once. We do not know from our data the context in which these sexual encounters took place: did it occur during sex work? out of social pressure to “fit in”? is their sexual identity fluid? This context is very important and it is missing from our data. The same is true for women who report being heterosexual yet have had sex with women. We were recently awarded a pilot grant to follow up with the WSWM subsample and ask them these kinds of questions. 

For the current paper, we see identity and behavior as separate, yet related, factors that may contribute to minority stress and negative health outcomes in different ways (similar to studies of the health of men who have sex with men (MSM) regardless of their sexual identity). We have added clarifying statements beginning in the Measures: Individual-level section to make this clearer for the reader. 

3. Please verify the name for 'substance dependence severity scale (SDS)' vs. 'Severity of Dependence Scale (SDS)' (line 188). It would be interesting if authors could explain the reason why they used more conservative cut-off scores as suggested by other researchers - e.g., were the populations similar to the one assessed?

We fixed the error with the scale title in the text and made some edits to the description of the measure. The scale as originally developed by the WHO uses the same items and cut-off scores for all types of drugs (e.g., cocaine, heroin, meth, marijuana, etc.). Other researchers have developed drug-specific cut-off scores by examining the scale properties specifically for meth users, heroin users, etc. We chose to use these more refined scores in order to be more precise and cautious in our analysis. 

4. In the discussion section, since it is a very particular population, it would be nice to have national rates of the individual-level, interpersonal-level, structural-level and health outcomes that are assessed, or at least identifying those that are rather different from the national level and actually citing the literature rate. For example, line 328 of the article states that the reported rates of childhood trauma, IPV, and incarceration history, overall, far exceed national rates - which are.... I believe this would further show the importance of the present findings and also of how relevant it is to study minority populations like these.

This is a great suggestion and we have done this.

---

## [Decision Letter · Decision Letter 1]

21 Feb 2020

PONE-D-19-19530R1

Documentation of disparities for women who have sex with women and men using a cohort of Mexican-American Women

PLOS ONE

Dear Dr. Cepeda,

Thank you for submitting your manuscript to PLOS ONE. After careful consideration, we feel that it has merit but does not fully meet PLOS ONE’s publication criteria as it currently stands. Therefore, we invite you to submit a revised version of the manuscript that addresses the points raised during the review process.

Provide specific feedback from your evaluation of the manuscript

We would appreciate receiving your revised manuscript by Apr 06 2020 11:59PM. To enhance the reproducibility of your results, we recommend that if applicable you deposit your laboratory protocols in protocols.io, where a protocol can be assigned its own identifier (DOI) such that it can be cited independently in the future. For instructions see: http://journals.plos.org/plosone/s/submission-guidelines#loc-laboratory-protocols

We look forward to receiving your revised manuscript.

Kind regards,

Geilson Lima Santana, M.D., Ph.D.

Academic Editor

PLOS ONE

Additional Editor Comments (if provided):

Dear authors, I will be glad to accept your submission for publication after some small changes suggested by one the reviewers:

"The authors have addressed all comments and the article has improved, but they have misunderstood my first comment regarding the title. While I wrote "The data shown cover so many different aspects that should NOT to be simplified as just the "documentation of disparities", they understood as if I was suggesting to actually use the terms "documentation of disparities". Besides this minor correction and provided a new title that encompasses the range of multi-level risk factors that were assessed in the present study".

Best wishes, Geilson.

Reviewers' comments:

Reviewer's Responses to Questions

**Comments to the Author**

1. If the authors have adequately addressed your comments raised in a previous round of review and you feel that this manuscript is now acceptable for publication, you may indicate that here to bypass the “Comments to the Author” section, enter your conflict of interest statement in the “Confidential to Editor” section, and submit your "Accept" recommendation.

Reviewer #2: All comments have been addressed

2. Is the manuscript technically sound, and do the data support the conclusions?

Reviewer #2: No

3. Has the statistical analysis been performed appropriately and rigorously? 

Reviewer #2: Yes

4. Have the authors made all data underlying the findings in their manuscript fully available?

Reviewer #2: No

5. Is the manuscript presented in an intelligible fashion and written in standard English?

Reviewer #2: Yes

6. Review Comments to the Author

Reviewer #2: The authors have addressed all comments and the article has improved, but they have misunderstood my first comment regarding the title. While I wrote "The data shown cover so many different aspects that should NOT to be simplified as just the "documentation of disparities", they understood as if I was suggesting to actually use the terms "documentation of disparities". Besides this minor correction and provided a new title that encompasses the range of multi-level risk factors that were assessed in the present study.

7. PLOS authors have the option to publish the peer review history of their article (what does this mean?). If published, this will include your full peer review and any attached files.

Reviewer #2: No

---

## [Author Response · Author response to Decision Letter 1]

25 Feb 2020

Additional Editor Comments (if provided):

Dear authors, I will be glad to accept your submission for publication after some small changes suggested by one the reviewers:

"The authors have addressed all comments and the article has improved, but they have misunderstood my first comment regarding the title. While I wrote "The data shown cover so many different aspects that should NOT to be simplified as just the "documentation of disparities", they understood as if I was suggesting to actually use the terms "documentation of disparities". Besides this minor correction and provided a new title that encompasses the range of multi-level risk factors that were assessed in the present study".

Best wishes, Geilson.

I see how this was a misunderstanding and am glad to have the opportunity to revise the title. The new title is, “Examination of Multilevel Domains of Minority Stress among Latina Women who have Sex with Women and Men.”

---

## [Editor Report · Decision Letter 2]

2 Mar 2020

Examination of Multilevel Domains of Minority Stress: Implications for Drug Use and Mental and Physical Health among Latina Women who have sex with Women and Men

PONE-D-19-19530R2

Dear Dr. Cepeda,

We are pleased to inform you that your manuscript has been judged scientifically suitable for publication and will be formally accepted for publication once it complies with all outstanding technical requirements.

With kind regards,

Geilson Lima Santana, M.D., Ph.D.

Academic Editor

PLOS ONE

---

## [Editor Report · Acceptance letter]

9 Mar 2020

PONE-D-19-19530R2 

Examination of Multilevel Domains of Minority Stress: Implications for Drug Use and Mental and Physical Health among Latina Women who have sex with Women and Men 

Dear Dr. Cepeda:

I am pleased to inform you that your manuscript has been deemed suitable for publication in PLOS ONE. Congratulations! Your manuscript is now with our production department. 

With kind regards,

on behalf of

Dr. Geilson Lima Santana 

Academic Editor

PLOS ONE